# 1 Application of Fengyun 3-C GNSS occulation sounder for

- assessing global ionospheric response to magnetic storm event
- Weihua Bai<sup>1,2,3</sup>, Guojun Wang<sup>1,4</sup>, Yueqiang Sun<sup>1,2,3</sup>, Jiankui Shi<sup>1,3,4</sup>, Xiangguang Meng<sup>1,2</sup>,
- Dongwei Wang<sup>1,2</sup>, Qifei Du<sup>1,2</sup>, Xianyi Wang<sup>1,2</sup>, Junming Xia<sup>1,2</sup>, Yuerong Cai<sup>1,2</sup>, Congliang
- Liu<sup>1,2</sup>, Wei Li<sup>1,2</sup>, Chunjun Wu<sup>1,2</sup>, Danyang Zhao<sup>1,2</sup>, Di Wu<sup>1,2</sup>, Cheng Liu<sup>1,2</sup>

<sup>1</sup>National Space Science Center, Chinese Academy of Sciences, Beijing 100190, China

<sup>2</sup>Beijing Key Laboratory of Space Environment, Beijing 100190, China

- <sup>3</sup>University of Chinese Academy of Sciences, Beijing 100049, China
- <sup>4</sup>State Key Laboratory of Space Weather, Beijing 100190, China
- Correspondence to: Guojun Wang (gjwang@nssc.ac.cn )

### 14 Abstract.

The rapid advancement of global navigation satellite system (GNSS) occultation technology in recent years has 16 made it one of the most advanced space detection technologies of the 21st century. GNSS radio occultation has 17 many advantages, including all-weather operation, global coverage, high vertical resolution, high precision, 18 long-term stability, and self-calibration. Data products from GNSS occultation sounding can greatly enhance 19 ionospheric observations and contribute to space weather monitoring, forecasting, modeling, and research. In this 20 study, GNSS occultation sounder (GNOS) results from a radio occultation sounding payload aboard the Fengyun 21 3-C (FY3-C) satellite were compared with ground-based ionosonde observations. Correlation coefficients for 22 peak electron density (NmF2) derived from GNOS Global Position System (GPS) and Beidou navigation system 23 (BDS) products with ionosonde data were higher than 0.9, and standard deviations were less than 20 %. Global 24 ionospheric effects of the strong magnetic storm event in March 2015 were analyzed using GNOS results 25 supported by ionosonde observations. The magnetic storm caused a significant disturbance in NmF2 and hmF2 26 levels. Suppressed daytime and nighttime NmF2 levels indicated mainly negative storm conditions. In the zone 27 of geomagnetic inclination between 40-80°, average NmF2 during the geomagnetic storm showed the same 28 basic trends in GNOS measurements, and in observations from 17 ground-based ionosonde stations, and 29 confirmed the negative effect of the event on the ionosphere. The analysis demonstrates the reliability of the 30 GNSS radio occultation sounding instrument GNOS aboard the FY3-C satellite, and confirms the utility of 31 ionosphere products from GNOS for statistical and event-specific ionospheric physical analyses. Future FY3

series satellites, and increasing numbers of Beidou navigation satellites, will provide increasing GNOS
 occultation data on the ionosphere, which will contribute to ionosphere research and forecasting applications.

### 3 1 Introduction

Global navigation satellite system (GNSS) occultation detection uses occultation receivers mounted on low earth orbit (LEO) satellites to collect GNSS signals that are refracted and delayed by the atmosphere and ionosphere. The excess phase due to the atmosphere and ionosphere is determined from measurements of the delayed carrier phase, and the precise positions and velocities of the LEO and GNSS satellites. An inverse Abel transform method is used to derive electron density of the ionosphere, the refraction index, temperature, humidity, and atmospheric pressure data, as shown in Fig. 1.

10 GNSS radio occultation technology makes global scale measurements of the atmosphere and ionosphere possible. 11 It has the advantages of high precision, high vertical resolution, long-term stability, global coverage, all-weather 12 operation, and a relatively low-cost, which can compensate for some of the shortcomings of conventional 13 atmospheric and ionospheric sounding tools (e.g., Fu et al., 2007). The global scale data obtained has important 14 scientific potential for improving the accuracy of numerical weather prediction, near space environment 15 monitoring research, global climate change research, atmospheric modeling research, and data assimilation. 16 Radio occultation technology has significant scientific value and a broad array of potential practical applications 17 in climatology, meteorology, ionospheric studies, and geodesy.

18 Radio occultation is extremely useful and important in ionosphere research, monitoring ionospheric anomalies, 19 investigating ionospheric scintillation, and forecasting space weather. It also has a broad range of potential 20 applications in communications, space operations, and national defense. Zhao et al. (2013) used ionosonde and 21 radio occultation data to analyze differences in the ionosphere in eastern and western China, including the 22 origins of ionospheric changes and latitudinal and longitudinal changes in structural layers of the ionosphere. Liu 23 et al. (2008, 2009, 2010, 2011) used COSMIC radio occultation data to study seasonal changes in the electron 24 density of the ionosphere, characteristics of the low latitude ionosphere, and the scale height of the ionospheric 25 peak. In addition, many researchers have used ionospheric occultation data to search for ionospheric anomalies 26 prior to earthquakes (Yang, et al., 2008; Zhang, et al., 2008). As well as using GNSS radio occultation data to 27 obtain ionospheric electron density profiles, new and innovative exploratory studies continue to emerge; for 28 example, using amplitude and signal-to-noise ratio data from radio occultation to detect the Es layer (Hocke, et 29 al., 2001), and spread F (Lu, et al., 2011).

30 The GNSS radio occultation sounder (GNOS) instrument (Fig. 2), developed by the National Space Science 31 Center of the Chinese Academy of Sciences (NSSC, CAS), has accumulated large amounts of radio occultation 32 data since it was launched into orbit aboard the Fengyun 3 C (FY3-C) satellite on September 23, 2013. It was the 33 first GNOS instrument compatible with both Beidou navigation satellite system (BDS) and Global Positioning 34 System (GPS) technology. FY3-C is in a sun-synchronous polar orbit, at an altitude of 836 km, inclination of 35 98.8°, and has an orbital period of 101.5 minutes. The atmospheric refractivity profile of GNOS has a precision 36 of less than 1 % in 5-25 km (Bai, et al., 2014; Liao, et al., 2016; Wang, et al., 2015). Peak electron density in the 37 ionosphere can be detected to within 20 % of ionosonde measurements (Wang, et al., 2015; Yang, 2015).

The purpose of this study is to explore applications of FY3-C GNOS ionospheric data products in space weather

- research, specifically in the analysis of ionospheric NmF2 patterns during the magnetic storm of March 2015.
- Results of the study lay the foundation for the use of GNOS in space weather research, including magnetic and
- ionospheric storms.

# 5 2 Instrument performance and data validation

The GNOS aboard the FY3-C satellite is composed of two fixed occultation antennas, one forward and one aft. 7 The electronic unit is located in the cabin. The forward and aft occultation antennae are each electrically split 8 into atmospheric and ionospheric components (Bai, et al., 2014). The ionospheric occultation antennas are single 9 unit, micro-strip, dual-mode, and dual-frequency, and they can simultaneously receive BDS dual-frequency (B1 10 and B2) and GPS dual frequency (L1 and L2) ionospheric occultation signals. The maximum gain of each 11 antenna is 5 dB, and the half power beam width of the ionospheric occultation antenna is  $\pm 40^{\circ}$ . The forward 12 ionospheric occultation antenna is oriented normal to the +X axis of the satellite, i.e., the direction in which the 13 satellite is moving. The aft ionospheric occultation antenna is oriented normal to the -X axis of the satellite. The 14 objective of this design is to make the beam of the ionospheric occultation antenna with maximum gain cover the 15 ionospheric occultation target region, in order to obtain a high signal-to-noise-ratio (SNR), and high quality 16 occultation signal.

17 Within the power consumption limits aboard the FY3-C satellite, GNOS is equipped with six dual-frequency 18 GPS occultation channels, which are able to simultaneously track dual-frequency signals from six GPS satellites 19 (including atmospheric and ionospheric occultation signals). It also has four dual-frequency BDS occultation 20 channels, which can simultaneously track four dual-frequency BDS satellite signals (including atmospheric and 21 ionospheric occultation signals). The GNOS ionospheric occultation data mainly includes dual-system, dual-22 frequency carrier phase and SNR information, with a sampling rate of 1 Hz. Since the primary mission is 23 atmospheric occultation sounding, this is given priority, so that when there are no free channels, a new 24 atmospheric occultation event occupies an ionospheric occultation channel. Under these limited channel 25 conditions, the actual number of complete ionospheric occultation profiles that can be produced daily is around 26 220 from the GPS and around 130 from the BDS.

GNSS signals transmitted through the ionosphere and received by LEO satellites are bent and delayed by refraction in the ionosphere. Using dual-frequency phase observations, the corresponding total electron content (TEC) of the ionosphere can be obtained:

(1)

 $TEC = \frac{f_1^2 f_2^2}{C(f_1^2 - f_2^2)} (L_1 - L_2)$ 

where  $L_1$  and  $L_2$  are the dual-frequency carrier phase observations,  $C=40.3082 \text{ m}^3 \text{ s}^{-2}$  is a constant, and  $f_1$  and  $f_2$ indicate the two frequencies. This type of dual frequency TEC inversion method (Syndergaard, et al., 2000; Datta-Barua, et al., 2008) eliminates clock differences and other instrumental biases, and also allows information on bending angle and impact height to be obtained. Equation (2) uses an inverse Abel transformation to obtain the refraction index, then electron density in the ionosphere is derived from the A-H formula (Hartree, 1931; Appleton, 1932).

1 
$$n(a) = \exp\left(\frac{1}{\pi} \int_{a}^{\infty} \frac{\alpha(x)dx}{\sqrt{x^{2} - a^{2}}}\right)$$
 (2)

2 where *n* is the ionosphere refraction index,  $\alpha$  the bending angle, and *a* the impact height.

3 We evaluated the GNOS ionosphere electron density profiles through statistical comparison with ionosonde 4 data. Electron density profile data were collected from GNOS, covering a 365-day period between October 5 1, 2013 and September 30, 2014, giving 81,215 occultation profiles from the GPS and 50,356 from the 6 BDS. We also collected ionosphere observation data taken from ground ionosonde stations, which mainly 7 comprised two parameters: maximum electron density in the F2 layer of the ionosphere (NmF2); and the 8 altitude of the maximum electron density (hmF2). Ionosonde data were obtained from 69 ionosonde 9 stations of the U.S. Space Weather Prediction Center (SWPC) (as shown in Fig. 3). The criteria used for 10 matching GNOS and ionosonde data were a time interval within ±1 hour, and geographic latitude and longitude 11 within  $\pm 2^{\circ}$ . For every matching pair of data, the relative error (R) in the GNOS NmF2 was calculated using Eq. 12 (3):

(3)

14 where the subscript *IONO* represents ionosonde.

 $R = \frac{NmF2_{GNOS} - NmF2_{IONO}}{NmF2_{IONO}} \times 100\%$ 

15 Figure 4 compares NmF2 measurements from the GNOS GPS occultation and the ionosondes. Over the 16 course of the year, a total of 547 matching pairs of data were collected. Since the FY3-C satellite is in a 17 sun-synchronous polar orbit, it passes ground stations at around 10:00 and 22:00 (LT), hence, occultation 18 events are mainly concentrated in the two local time periods between 09:00-12:00 and 21:00-24:00. Linear 19 regression of absolute NmF2 values derived from each methods (Fig. 4), gives a correlation coefficient of 20 0.95, statistical bias of 3.00 %, and standard deviation of 17.93 %. Figure 5 is similar to Fig. 4, but with 21 GNOS BDS occultation profiles rather than GPS products, and shows most occultation events also occurred at 22 local times of 9:00-12:00 and 21:00-24:00. The correlation coefficient of the fitted regression is 0.95, statistical 23 bias is 4.67 %, and standard deviation is 19.19 %.

24 Global electron density profiles have been successfully probed in several previous GNSS radio occultation 25 missions, including GPS/MET, CHAMP, and COSMIC. Using ionosonde data for verification, their reported 26 precisions are: NmF2 average bias 1 %, and standard deviation 20 % for GPS/MET (Hajj et al., 1998); NmF2 27 average bias -1.7 %, and standard deviation 17.8 % for CHAMP (Jakowski et al., 2002); insignificant mean 28 differences for COSMIC, with 22-30 % for NmF2 and 10-15 % for hmF2 (Limberger, et al., 2015). GNOS GPS 29 and BDS occultation NmF2 observations reported in this study have a slightly higher average bias than the other 30 systems, but their good correlation coefficients and standard deviations demonstrate the overall reliability of the 31 results.

#### 1 3 Analysis of GNOS results during the magnetic storm of March 2015

### 2 3.1 Characteristics of the magnetic storm

3 Solar activity in 2015 was at a moderate level, and there were several large geomagnetic storm events. This 4 study focuses on the magnetic storm event that occurred between March 11 and March 31, 2015, peaking at 5 05:22 UT on March 17. Changes in magnetic indices during the storm are shown in Fig. 6. The geomagnetic 6 activity index Kp, which characterizes global geomagnetic activity, was generally below 3 before March 17, then 7 it increased significantly, with large perturbations continuing until March 26, after which it returned to pre-storm 8 levels. The Dst index, which is an index of geomagnetic activity used to assess the severity of magnetic storms, 9 was relatively stable before March 17, hovering around zero. The index suddenly increased between 04:00-10 06:00 UT on the 17<sup>th</sup>, marking the initial phase of the magnetic storm, and then dropped rapidly during the main 11 phase to a minimum of -223 nT at 23:00 UT. A rapid recovery phase followed, from 23:00 UT on March 17 12 until 18:00 UT on March 18, then a slower recovery until around 10:00 UT on March 25, when it returned to 13 pre-storm levels. The auroral electrojet (AE) index mainly reflects polar substorm intensity, with variations 14 closely related to the quantity of particles injected into polar regions. The AL and AU indices reflect westward 15 and eastward electrojet conditions, and the AE index is the absolute difference between them. During the main 16 phase of the magnetic storm, the magnitude of AE index perturbations reached around 1500, with frequent lower 17 magnitude perturbations during the recovery phase. Overall, this magnetic storm event caused severe 18 geomagnetic disturbances on a global scale. As plasma in the ionosphere is controlled by the earth's magnetic 19 field, the global ionosphere was also affected by the magnetic storm.

### 20 3.2 Global GNOS results during the magnetic storm

21 Figure 7 shows global daytime (07:00-17:00 LT, Fig. 7a) and nighttime (19:00-05:00 LT, Fig. 7b) occultation 22 event distributions for March 14, 2015 (pre-storm calm), March 17 (main phase), March 18 (rapid recovery 23 phase), and March 22 (recovery phase). Around 350 ionosphere occultation profiles were recorded daily (220 24 GPS + 130 BDS). Although the quantity of data is limited due to the rather high inclination polar orbit of FY3-C, 25 recorded occultation events are distributed across the globe, with a relative concentration at higher latitudes. Fig. 26 7a shows that, before the magnetic storm, the highest daytime NmF2 values were mostly distributed around the 27 magnetic equator, with relatively low electron densities in high magnetic latitudes. During the magnetic storm, 28 NmF2 perturbations were significant, with especially large increases in the South Atlantic region. In the 29 southeastern Pacific, NmF2 decreased during the main storm phase, and increased during the recovery phase, 30 returning to pre-storm levels by March 22. Nighttime GNOS soundings (Fig. 7b) show suppressed NmF2 values 31 in East Asia and Australia during the main phase and start of the recovery phase of the storm. NmF2 values had 32 returned to pre-storm levels by March 22. These results demonstrate the capability of FY3-C GNOS for 33 characterizing global NmF2 response to magnetic storm events.

#### 34 3.3 Statistical analysis of GNOS ionosphere products during the magnetic storm period

The daily GNOS ionosphere profiles are relatively sparse, and unevenly distributed around the globe. Therefore, it is difficult to quantitatively analyze the response of a specific location during a magnetic storm event. However, as most occultation events are distributed in mid and high latitudes, it is possible to analyze changes in average NmF2 and hmF2 values during a magnetic storm. Figure 8 plots changes in average daytime and

- 1 nighttime NmF2 and hmF2 values as determined by GNOS, in the zone of geomagnetic inclinations between
- 2 40-80°, in both the northern and southern hemispheres. Samples from geomagnetic inclinations outside this zone
- 3 were excluded as there were insufficient radio occultation events for statistical analysis. Figure 8 shows the
- 4 following trends:
- (1) Nighttime average NmF2 levels were much lower in the main phase (March 17) and at the start of the
   recovery phase (March 18–20) than before the storm, reaching a minimum on March 18.
- 7 (2) In the northern hemisphere, nighttime average hmF2 increased rapidly during the main phase of the storm,
- 8 reaching a maximum on March 17, decreased to a minimum on March 20, and slowly increased again after.
- 9 However, in the southern hemisphere, average hmF2 was only slightly higher on the 17<sup>th</sup> than on the 16<sup>th</sup>. It
- 10 reached a minimum on the 18<sup>th</sup>, after which it slowly rose again. This pattern is essentially in accord with the 11 NmF2 trends.
- (3) Daytime average NmF2 in the northern hemisphere reached a maximum during the main phase of the
   magnetic storm on March 17, falling to a minimum on the 19<sup>th</sup>, after which it slowly increased. In the southern
   hemisphere, daytime average NmF2 began to fall on the 15<sup>th</sup>, reached a minimum on the 18<sup>th</sup>, and then slowly
   recovered.
- (4) Daytime average hmF2 results were similar for both the northern and southern hemispheres, reaching a
   maximum on March 17, and rapidly dropping to a minimum on the 18<sup>th</sup>, before gradually recovering.
- Overall, the magnetic storm caused significant disturbances in global NmF2 and hmF2 values in the region of magnetic inclinations between 40–80°, in both the northern and southern hemispheres. Both daytime and nighttime NmF2 values showed mainly negative storm characteristics, while hmF2 increased significantly during the main phase of the storm, and was suppressed at the start of the recovery phase.
- 22 3.4 Comparisons with ionosonde observations

23 In previous sections, analysis of the effects of the magnetic storm event on FY3-C satellite GNOS results 24 showed that it caused significant perturbations in global NmF2 levels. In this section, we compare GNOS 25 measurements with ionosonde data from the SWPC worldwide stations. As there are very few ionosonde stations 26 located at magnetic inclinations of 40-80° in the southern hemisphere, we focused on NmF2 data from the 15 27 ground stations in the northern hemisphere, in the period March 14-22, 2015. The geographic distribution of 28 these 15 stations is shown in Fig. 9. We also included NmF2 data from two China Meridian Project stations in 29 the analysis (shown as red stars in Fig. 9). Figures 10 and 11 plot NmF2 variations at each station. Perhaps the 30 most outstanding feature is a surge in NmF2 values at individual stations on March 17, during the main phase of 31 the magnetic storm (e.g., around 12:00 UT at Moscow), with a decrease after 16:00 at many stations. Almost all 32 the stations show a significant decrease in NmF2 measurements during the beginning of the recovery phase, on 33 March 18 and 19, except for a few western European stations (e.g., Rome). After March 19, NmF2 patterns 34 become more complex, with some stations (e.g., Mohe) showing values below pre-storm levels, and others (e.g., 35 Okinawa) about the same as pre-storm levels. Overall, ionospheric perturbations during the magnetic storm were 36 mainly negative.

Figure 12 compares average NmF2 values from the 17 ionosonde stations with those from the GNOS occultation products for different time periods in the magnetic storm event, including all day (24 h), 9:00–12:00 LT, and 21:00–24:00 LT (in Figs. 4 and 5 most of the GNON occultation events were concentrated around these times). In these plots, NmF2 trends are very similar for the two observation methods, with the negative storm effects of the recovery phase quite clear. However, , GNOS measurements, especially in the 21:00–24:00 time block, show larger perturbations than those of the ionosonde stations, indicating significant differences still exist between the two measurement techniques.

### 8 4 Discussion

9 The GNSS radio occultation allows monitoring of electron density in the ionosphere at a global scale. It has the 10 advantages of high accuracy, good vertical resolution, global coverage, and all-weather capability. However, an 11 important constraint in applications of the occultation electron density products is the assumption of a 12 symmetrical ionosphere in the inverse Abel transformation calculations. In reality, it is very difficult to guarantee 13 symmetrical distribution of electron density in the ionosphere, especially near anomalies at the magnetic equator. 14 Nevertheless, comparison of the GNOS probe results with ionosonde measurements provided a correlation 15 coefficient of 0.95, and standard deviation less than 20 %. Therefore, in the majority of cases, GNOS results may 16 be considered to be reliable and reasonable.

17 The study analyzed the effect of the March 2015 magnetic storm on the global ionosphere, using data from FY3-18 C GNOS and from ground-based ionosonde stations. In terms of spatial distribution, Figs. 7a and 7b showed that 19 daytime NmF2 values in the southern Atlantic region were elevated throughout the course of the storm. In the 20 southeastern Pacific, NmF2 values first decreased at the beginning of the main phase of the magnetic storm, and 21 then increased in the recovery phase. Nighttime NmF2 values around East Asia and Australia were mainly 22 suppressed during the main phase, and at the start of the recovery phase, and returned to pre-storm levels by 23 March 22. In terms of different geomagnetic inclination zones (Fig. 8), the magnetic storm had a significant 24 effect on NmF2 and hmF2 values. Both daytime and nighttime average NmF2 values showed negative storm 25 effects. It was not possible to perform long-term continuous comparison between ionosonde and GNOS data, as 26 the number of occultation events recorded by GNOS was insufficient. Hence, the results from two geomagnetic 27 inclination zones were averaged to enable a quantitative comparison. Besides, the statistical significance of this 28 method is based on the assumption that the effects of the magnetic storm are basically consistent throughout the 29 ionosphere in each magnetic latitudinal zone. The statistical comparison of the averaged data showed similar 30 general trends in measurements from the 17 ionosonde stations and GNOS during the March 2015 magnetic 31 storm. The comparison further confirmed the nature of the magnetic storm, and the negative effects of the storm 32 on the ionosphere.

The way in which the global ionosphere responds to magnetic storms is extremely complicated. From Fig. 7a, we can see that this particular magnetic storm caused varying responses in the ionosphere at different times and locations. Many physical factors influence the ionosphere, such as electric fields, and neutral winds. For a specific magnetic storm, corresponding ionospheric perturbations also depend on the season, solar activity, local time of magnetic storm occurrence, and the latitude and longitude. Therefore, ionospheric storms are extremely complex; no two storms are precisely alike, and the mechanisms that generate them also vary (Balan, et al., 1990;

Fuller-Rowell, et al., 1996; Danilov, et al., 2001; Mendilo, et al., 2006). In addition, the type and form of a
 magnetic storm also makes a difference in the way that it affects the ionosphere (Zhang, et al., 1995). Future
 analysis incorporating assimilation with other data sources and models may allow the precise mechanisms
 responsible for ionospheric effects of the March 2015 magnetic storm to be determined.

# 5 5 Conclusions

6 The comparison of ionosonde data and FY3-C GNOS radio occultation products presented in this study shows 7 that, in the majority of cases, GNOS data is reliable and reasonable. Based on ionosphere data from the FY3-C 8 GNOS payload combined with those from ground-based ionosonde, this study analyzed the characteristics of 9 global ionosphere response to the magnetic storm event in March 2015. Daytime NmF2 values increased in the 10 South Atlantic region, and first decreased and then increased in the southeast Pacific region. In East Asia and 11 Australia, nighttime NmF2 values were mainly suppressed, but recovered to pre-storm levels around March 22. 12 In the region of higher magnetic inclinations, NmF2 and hmF2 levels were clearly affected by the storm. 13 Daytime and nighttime NmF2 levels mainly indicated a negative storm response. Overall, the trend detected by 14 GNOS during the magnetic storm event in the zone of magnetic inclinations between 40-80° in the northern 15 hemisphere was similar to the trend detected by the 17 ground-based ionosonde stations in that region. This 16 further confirmed the negative response of the ionosphere to the March 2015 magnetic storm event. The study 17 demonstrates the reliability of FY3-C GNOS radio occultation measurements for analyzing statistical and event-18 specific physical characteristics of the ionosphere. More Beidou navigation satellites, and other FY3 series 19 satellites (FY3-D, E, F, G, and H), are planned in the future, and their GNOS payloads offer the potential for 20 generation of significantly more data in support of ionospheric physics research and forecasting applications.

# 21 Acknowledgments

We thank UML GIRO for providing the ionosonde data. We also acknowledge the use of data from the Chinese
 meridian Project. This research was supported jointly by National Science Fund (41505030, 41405039,
 41405040 and 41574150 and 41606206) and Scientific Research Project of Chinese Academy of Sciences
 (YZ201129).

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
