# Peer review of "Application of Fengyun 3-C GNSS occulation sounder for"

_Atmospheric Measurement Techniques, 2016_

## Referee Comment (RC1) · Anonymous Referee #1 · 28 Dec 2016

This paper reports on retrieval of NmF2 (peak electron density) derived from GNOS (combined GPS + Beidou navigation products). It compares the retrieved NmF2 to ionosonde data. Fengyun is in a sun-synchrous orbit at 836 km, with an inclination of 98.8 degrees. GNOS can track up to 6 GPS satellites and 4 Beidou satellites at any given instant. Daily retrievals are about 220 from GPS and 130 from Beidou. Eqn. 1 is stated to eliminate clock differences and other instrumental biases.

365 day period between oct 1,2013 and sep 20, 2014 used. Figure 4 compares NmF2 measurements from the GNOS GPS occultation and Figure 5 shows those produced by GNOS BDS occultation.

The comparison between these two figures is the only thing I find in this paper that

could be called a "new finding." The differences between these two figures should be discussed in detatil.

What is new and different about this paper. I see that 1) it provides data from the Fengyun 3-C occultation sensor, and that this satellite is adequately described in the paper. 2) it is using a combined Beidou and GPS sensor.

1) The authors do not adequately describe the Beidou constellation, signals or frequencies. Beidou is still a relatively new system and has not been well-referenced in the literature yet. Why are the results between Beidou and GPS different? The authors should discuss this in detail and provide more background introduction to Beidou. 2) Not enough background material was provided. In particular, the paper by Hararulema and Carelse (2016). Their paper was published before yours and discusses the first long-term comparison between RO and ionosonde NmF2 and hmF2 data during storm conditions. THey also provide results (a similiar finding to yours) that NmF2 and hmF2 agree to within 21% and 9% (1 standard deviation), respectively. They also saw that maximum deviations for both NmF2 and hmF2 occur during high solar activity periods 3) I do not think that equation 1) eliminates all the biases. There are both satellite and receiver biases (differential delay differences between the two frequencies) that are not eliminated in this equation. Please discuss. Add or discuss these references in the paper. Reference: John Bosco Habarulema and Suné Arlene Carelse, (2016) Long-term analysis between radio occultation and ionosonde peak electron density and height during geomagnetic storms NmF2, HmF2 4110 Geophysical Research Letters 10.1002/2016GL068944

Other papers which should be mentioned are also given here: Habarulema, J. B., Z. T. Katamzi, and E. Yizengaw (2014), A simultaneous study of ionospheric parameters derived from FORMOSAT-3/COSMIC, GRACE, and CHAMP missions over middle, low, and equatorial latitudes: Comparison with ionosonde data, J. Geophys. Res. Space Physics, 119, 7732–7744, doi:10.1002/2014JA020192.

[Figure]

Garcia-Fernandez, M., M. Hernandez-Pajares, J. M. Juan, and J. Sanz (2003), Improvement of ionospheric electron density estimation with GPSMET occultations using Abel inversion and VTEC information, J. Geophys. Res., 108(A9), 1338, doi:10.1029/2003JA009952. Yue, X., W. S. Schreiner, J. Lei, S. V. Sokolovskiy, C. Rocken, D. C. Hunt, and Y.-H. Kuo (2010), Error analysis of Abel retrieved electron density profiles from radio occultation measurements, Ann. Geophys., 28, 217–222.

Yue, X., W. S. Schreiner, C. Rocken, and Y. H. Kuo (2011), Evaluation of the orbit altitude electron density estimation and its effect on the Abel inversion from radio occultation measurements, Radio Sci., 46, RS1013, doi:10.1029/2010RS004514.

---

## Author Comment (AC1) · 3 Jan 2017

Dear Editor and Referee #1,

Thank you for your comments concerning our manuscript entitled "Application of Fengyun 3-C GNSS occulation sounder for assessing global ionospheric response to magnetic storm event" (Manuscript Number: doi:10.5194/amt-2016-291-RC1). Those comments are all valuable and very helpful for revising and improving our paper, as well as the important guiding significance to our researches. We have studied comments carefully and have made correction which we hope meet with approval. The responds to the comments are as flowing:

[Figure]

Referee #1:

General comments:

1, Figure 4 compares NmF2 measurements from the GNOS GPS occultation and Figure 5 shows those produced by GNOS BDS occultation. The comparison between these two figures is the only thing I find in this paper that could be called a "new finding."

Response: We would like to explain our purposes and meanings in this paper: (1) It's a first demonstration for the application of the FY3-C GNSS occulation sounder (GNOS) for assessing global ionospheric response to magnetic storm events. These results coincide with previous studies (e. g. Habarulema et al. 2014, 2016and references therein), which just right proves the reliability of FY3-C GNOS radio occultation measurements for analyzing statistical and event-specific physical characteristics of the ionosphere. (2) Using the one year GNOS data, we showed that the Correlation coefficients NmF2 derived from GNOS GPS and BDS products with ionosonde data were higher than 0.9, and standard deviations were less than 20 %. It's important for the Multi-GNSS occultation application, which proves the precisions of the different GNSS occultations are consistent and comparable. (3) We analyzed the variation of the daily and zonal average of the NmF2 and hmF2 values during the whole geomagnetic storm, and confirmed the negative effect of this space weather event on the ionosphere. In the zone of geomagnetic inclination between 40–80°, average NmF2 during the geomagnetic storm showed the same basic trends in GNOS measurements, and in observations from 17 ground-based ionosonde stations (4) We believe that our study makes a significant contribution to the literature because this paper is just a little case and first step for the application of the GNOS. As a new member of the family of the occultation missions, the data of the GNOS will have significant potential application in space weather monitoring and forecasting, as well as modeling and research in the future, especially, with datum from other GNSS occultation missions.

2, The differences between these two figures should be discussed in details.

Response: Well, we have added some discussion in this part. For the description of the two figures, we think that it's adequate in original manuscript, we complement the NmF2 errors comparison between GPS and BDS "The bias and standard deviation of the NmF2 derived from the GNOS BDS occultation and GPS occultation are consistent and comparable. However, the bias and standard deviation of the NmF2 from BDS are slightly larger, it could be caused by the larger position errors of the BDS satellites, especially for the GEO satellites, and the different distribution of the occultation events". In my opinion, the reason of why BDS is worse than GPS is because of the larger position errors of the BDS satellites, because the BDS consists of three types of the navigation satellites, GEO, IGSO, and MEO, it's impossible to get the precise ( cm level ) orbit determination for the GEO satellites, usually, the errors of the GEO are meter level (in Lou Y., et al. 2016, Fig.7). In figure 5, the statistic results include all type of BDS satellites. Moreover, the distribution of the BDS occultation events is different, you can find out some information from the reference Fig2 of the Liao M. et al. 2016, where we can see most of the GEO occultation events appeared on the particular high latitude region, the IGSO events concentrated on the edge of two big circle, and MEO events were evenly distribution. Different location distribution should be different error feature. Therefore, the reason would be very complicated, considering the length of this paper and our purpose (to discuss the ionospheric response to magnetic storm event using the GNOS ionosphere products), we would not like to discuss more in this paper, but we have planed to study it using more BDS products (two years).

Reference: Liao, M., Zhang, P., Yang, G. L., Bi, Y. M., Y., Liu, Bai, W. H., Meng, X. G., Du, Q. F., and Sun, Y. Q.: Preliminary validation of refractivity from a new radio occultation sounder GNOS/FY-3C. AMT, 9: 781-792, 2016. Lou Y., Liu Y., Shi C., Wang B., Yao X., Zheng F.,: Precise orbit determination of BeiDou constellation: method comparison. GPS Solut., 20:259–268, 2016.

Specific comments:

1, The authors do not adequately describe the Beidou constellation, signals or frequencies. Beidou is still a relatively new system and has not been well-referenced in the literature yet. Why are the results between Beidou and GPS different? The authors should discuss this in detail and provide more background introduction to Beidou.

Response: Yes, it's a good suggestion, we have added the basic information of the BDS in the third paragraph of the introduction section in upload manuscript: "BDS is China's global navigation satellite system, which can provide coverage in the Asia-Pacific region with 5 geostationary orbit (GEO) satellites 5 inclined geosynchronous orbit (IGSO) satellites and 5 medium earth orbit (MEO) orbit satellites, currently (China satellite navigation office, 2016)", and gave the BDS offical document (BeiDou document,2016) as a new reference. We also complement the dual-frequencies information in the second paragraph in section 2 : "The ionospheric occultation antennas are single unit, micro-strip, dual-mode, and dual-frequency, and they can simultaneously receive BDS dual-frequency (B1I 1561.098MHz and B2I 1207.140MHz) and GPS dual frequency (L1 1575.420MHz and L2 1227.600MHz) ionospheric occultation signals."

Reference: BeiDou navigation satellite system signal in space interface control document open service signal (Version 2.1): http://en.beidou.gov.cn/, last access: November, 2016.

2, Not enough background material was provided. In particular, the paper by Hararulema and Carelse (2016). Their paper was published before yours and discusses the first long-term comparison between RO and ionosonde NmF2 and hmF2 data during storm conditions. They also provide results (a similiar finding to yours) that NmF2 and hmF2 agree to within 21% and 9% (1 standard deviation), respectively. They also saw that maximum deviations for both NmF2 and hmF2 occur during high solar activity periods

Response: Thank you for your good suggestions. We have supplemented some background material in introduction and other necessary sections. Please see them in the

fourth paragraph in the last paragraph of the section 2, and the first paragraph in section 4. We would like to complement the differences between the paper Hararulema and Carelse (2016) and our manuscript. For example, the data we used is in the zone of geomagnetic inclination between 40–80° in the north hemisphere, including the GNOS ionosphere products (GPS and BDS) and the observations from 17 ground-based ionosonde stations; the ionosonde data of the paper Hararulema and Carelse (2016) is just one station in Grahamstown, south Africa, in the south hemisphere. Moreover, in our paper, we discuss the daily changing process of the Nmf2 and hmf2 during the whole geomagnetic storm. We showed the variation of the daily, daytime, and nighttime average of the NmF2 and hmF2 in the zone of geomagnetic inclination between 40–80° (in Fig.8 and Fig.12). Finally, we confirmed the negative effect of this geomagnetic storm event on the ionosphere.

3, I do not think that equation 1) eliminates all the biases. There are both satellite and receiver biases (differential delay differences between the two frequencies) that are not eliminated in this equation. Please discuss. Add or discuss these references in the paper.

Response: Yes, the referrer is right, the equation (1) can not eliminates all the biases, but, in original manuscript (in the fourth paragraph of the section 2) : "This type of dual frequency TEC inversion method (Syndergaard, et al., 2000; Datta-Barua, et al., 2008) eliminates clock differences and other instrumental biases, and also allows information on bending angle and impact height to be obtained", what we said is "other instrumental biases". In our opinion, the instrumental biases mean the biases or delay caused by the instrument (or payload or receiver system). The receiver clock offset and differences belong to the instrumental biases, and "other instrumental biases" includes the delay cause by cable connecting the RO antenna and receiver, and the delays or biases caused by the electronic components in the receiver (e.g. amplifiers, filters, and mixers). All of the instrumental biases could be considered the same for both two frequencies (L1,L2 of GPS, and B1I,B2I of BDS). I'm sorry for the expression

" instrumental biases" might be cause the misunderstanding, so we revised this sentence to "This type of dual frequency TEC inversion method (Syndergaard, et al., 2000; Schreiner, et al., 1999) eliminates clock errors (Jin S. et al, 2014), and also allows information on bending angle and impact height to be obtained", and we also added a reference (Jin S. et al, 2014), in which : P114, "The phase difference cancels out the orbit and clock errors automatically and . . . . . . . . ."

Reference: Jin S., Estel C., Xie F. (Eds.): GNSS remote sensing theory, methods and applications, EARSel Series, Springer, Freek D., Netherlands, 2014.

4, Others mentioned papers:

Thanks for the referee to supply the valuable references, we studied them and appended them in the suitable place. Garcia-Fernandez, et al., 2003, Habarulema, et al., 2014 and 2016 appended in the last paragraph of the section 2 to enhance the background introduction. Yue, et al., 2010, 2011 have been used to explain the Abel inversion in the first paragraph in section 4.

We appreciate for your warm work earnestly, and hope that the response will meet with approval.

The references and the new update version paper are enclosed in supplement.

Thank you very much.

Yours sincerely,

Bai Weihua, Wang Guojun and orther co-authors

Please also note the supplement to this comment:
http://www.atmos-meas-tech-discuss.net/amt-2016-291/amt-2016-291-AC1-supplement.zip

---

## Referee Comment (RC2) · 22 Apr 2017

**Referee comments – Chris Watson**

Manuscript title: Application of Fengyun 3-C GNSS occultation sounder for assessing global ionospheric response to magnetic storm event

Authors: Bai et al.

**General Comments:**

This manuscript discusses ionospheric radio occultation measurements of the GNSS receiver (GNOS) onboard the sun-synchronous FY3-C satellite launched Sept. 23, 2013. Validation of GNOS radio occultation (RO) derived NmF2 with ground-based ionosonde measurements is presented for a one year period. Global variations in RO NmF2, hmF2 and ionosonde NmF2 during the March 2015 geomagnetic storm are also presented.

The primary findings are that GNOS provides reliable ionospheric NmF2, and that GNOS measurements can be used to observe the average trend of ionospheric NmF2 and hmF2 associated with a geomagnetic storm at mid to high latitudes. The validation of NmF2 is fine, however I have concerns with the results based on averaging of the occultation-derived NmF2, and the characterization of the ionosphere in localized regions based on this averaging. The concerns along with other specific comments are itemized below.

The ionospheric response to the March 2015 storm has been extensively studied using (e.g. Astafyeva et al. 2015, Nava et al., 2016, etc.. a quick Google search reveals many). Discussion and references to these previous studies should be added, as well as consistencies or inconsistencies between GNOS RO observations and pervious results. Also add discussion on *Habarulema et al.,* 2016, "Long-term analysis between radio occultation and ionosonde peak electron density and height during geomagnetic storms", which is directly related to the analysis attempted in your study.

Also, GNOS occultation measurements are a valuable contribution to existing RO constellations. It would be worthwhile to add some discussion highlighting the uniqueness of GNOS RO measurements in terms of existing RO capabilities, and the specific RO studies that the high elevation, sun-synchronous FY3-C orbit may allow for.

**Specific Comments:**

1. P2 L30-37 Please specify receiver and antenna models.

2. Figure 2: Individual hardware components shown in the figure should be labelled.

3. P3 L33: Equation 1 does not eliminate differential code biases due to receiver and satellite hardware, as implied in the text. Please discuss the techniques applied to account for these biases.

4. Equation 2: Direct inversion of the TEC is usually sufficient for obtaining F region ionospheric densities. Is there a reason for using the bending angle inversion?

5. Equation 2: Please provide the method used for obtaining bending angle from the excess phase. Also specify how bending angles above satellite altitude are accounted for (since you are integrating to infinity).

6. Please indicate whether ionograms were scaled manually or "auto-scaled". Ionospheric parameters derived from manually scaled ionograms are generally more reliable.

7. State the maximum tangent point – ionosonde separation distance used in NmF2 validations.

8. Since occultation hmf2 is being used for analysis in this study, it should be validated as well. It shouldn't be too difficult to compare occultation and ionosonde hmf2, similar to the Nmf2 validation.

9. The Discussion in Section 3.2 seems to imply that the variations in NmF2 are a geographical effect (e.g. Line 28: "large increases in the South Atlantic region"), however these are observations over a 10 hour period, and thus temporal effects would be large, particularly during a geomagnetic storm. I'm not convinced that a few occultation events over a 10 hour period are sufficient to characterize the ionospheric behavior in a particular geographical region. From Figures 7a-b, the most I would conclude is that equatorial/low latitude NmF2 increases during daytime, and decreases at night. I have the same concerns for the geographical trends are also discussed on Page 7, Lines 17-23.

10. Figure 8 averages NmF2/hmF2 from mid-latitude, trough, and auroral regions. Since the ionospheric structure can vary significantly over these regions, please comment on the potential effects of this averaging, and whether the trends shown in Figure 8 would change if only mid-latitudes or auroral regions were considered.

11. On a related note, please comment on the occurrence magnetospheric substorm activity, which can result in significantly enhanced ionization in the nighttime auroral region. I would strongly suggest analyzing mid-latitude and auroral regions separately, instead of a broad region covering 40 to 80 degrees inclination.

12. The standard deviation for each averaged HmF2 and NmF2 value should be shown in Figures 8 and 12, perhaps as error bars.

13. P6 L8: NmF2 is maximum on March 16 according to Figure 8, as opposed to March 17 as stated in the text.

14. P6 L32: Is there an explanation for the few stations that observed a sustained daytime NmF2 enhancement on March 18?

15. Integrating NmF2 over all local times in the top panel of Figure 12 seems meaningless.

16. P7 L5-6: GNOS observations at 40-80 magnetic inclination extend into the auroral region, well poleward of the northernmost ionosonde station (Moscow) in Figure 9, and thus the averaged ionosonde NmF2 wouldn't include significant auroral region effects. This may help explain discrepancies in Figure 12.  For completeness, consider including ionosonde measurements from stations north of Moscow, of which there are several.

17. P7 L6-7: "..indicating significant differences still exist between the two measurement techniques."  Please specify the differences this statement is referring to.

18. P7 L13:  Spherical homogeneity of ionospheric density is also a very large assumption at high latitudes (trough, auroral, polar cap regions).

19. P8 L7:  Instead of "GNOS data", specify that GNOS NmF2 values are reliable, since this was the only parameter validated in the manuscript.

**Technical Corrections:**

P1 L16: I suggest "space detection" be replaced with "space-based remote sensing" or something along those lines.

P1 L26-28:  Unless I'm misinterpreting the intended meaning, this sentence should read something like: "In the zone of 40-80° magnetic inclination, average NmF2 observed by GNOS and 17 ground-based ionosondes showed the same basic trends during the geomagnetic storm.

P2 L4: "The GNSS radio occultation technique uses occultation receivers…"

Figure 1 axis labels are difficult to read.

---

## Author Comment (AC2) · 10 May 2017

Dear Chris Watson and Editor

Thank you for your comments concerning our manuscript entitled "Application of Fengyun 3-C GNSS occulation sounder for assessing global ionospheric response to magnetic storm event" (Manuscript Number: doi:10.5194/amt-2016-291-RC1). The comments and revises are very important for our paper's improving. We discussed and studied those valuable comments earnestly and made the updated manuscript and response carefully, we hope meet with approval. The new manuscript in the supplement is based on the helpful comments from the two referees during the state of the open discussing.

[Figure]

The responds to the nice comments from Chris are as flowing:

General comments: 1ãĂĄ ...... The ionospheric response to the March 2015 storm has been extensively studied using (e.g. Astafyeva et al. 2015, Nava et al., 2016, etc.. a quick Google search reveals many). Discussion and references to these previous studies should be added, as well as consistencies or inconsistencies between GNOS RO observations and pervious results. Also add discussion on Habarulema et al., 2016, "Long-term analysis between radio occultation and ionosonde peak electron density and height during geomagnetic storms", which is directly related to the analysis attempted in your study. ......

Response:

Thanks for the very good suggestions. We have added some discussion including the previous studies that you suggested. Please see Para. 5 in Section 1, Para. 1-4 in Section 4, and also renewed the Reference list.

Specific Comments:

1. P2 L30-37 Please specify receiver and antenna models.

Response:

According to the comment, we have revised the introduction to the payload GNOS in the first paragraph of the section 2, Page 3, line 8-10.

2. Figure 2: Individual hardware components shown in the figure should be labelled.

Response:

We have labeled the individual components in Fig. 2 in the uploaded manuscript. Thanks.

3. P3 L33: Equation 1 does not eliminate differential code biases due to receiver and satellite hardware, as implied in the text. Please discuss the techniques applied to
account for these biases.

Response:

Right. The code biases due to receiver and satellite hardware are difficult to eliminate totally. For the receiver, in order to eliminate the hardware delay of the two frequencies for one occultation satellite, we finished the calibration using the simulator on the ground before the launching, which can make the hardware delay caused by receiver lower than 1 ns. For the delay of the GPS satellite, we can get the correcting parameter form the GPS service centre (e.g. IGS). In equation 1, the L1 and L2 imply the ambiguities of the carrier phases and the hardware biases, so the TEC is the related TEC. It's not the absolute TEC. Moreover, begin with the equation(1), the derivative of the TEC can directly obtain electron density (see in Eq.(14) of Schreiner, et al., 199; Eq. (7.16) in Jin S., et al., 2014), so we do not need to consider the effect of the constant and bias in one occultation event processing.

4. Equation 2: Direct inversion of the TEC is usually sufficient for obtaining F region ionospheric densities. Is there a reason for using the bending angle inversion?

Response:

According to our old inversion method, we derived the bending angle and impact height from TEC, bending angle , used able inversion to get refractivity, then according to , we could get the electron density , eventually. But, we had updated the better inversion method (see in Eq.(14) of Schreiner, et al., 199; Eq. (7.16) in Jin S., et al., 2014) to obtain electron density profiles directly. Therefore, here is a mistake. We have replaced the Eq.(2) to , where is Electron density, r0 is the straight-line impact distance, see in section 2, Page 4 , line 3-5 in new manuscript.

5. Equation 2: Please provide the method used for obtaining bending angle from the excess phase. Also specify how bending angles above satellite altitude are accounted for (since you are integrating to infinity).

Response:

The same as the comment 4, we have corrected the Eq.2, and don't need to derive the bending angle. It's a very good question for the upper limit of integral. Because the altitude of the FY3C is around 833km, it's a better height to integral; the effect of the integral from the LEO satellite altitude to infinity is small enough, and neglected.

6. Please indicate whether ionograms were scaled manually or "auto-scaled". Ionospheric parameters derived from manually scaled ionograms are generally more reliable.

Response:

we downloaded the ionosonder products of the SWPC worldwide stations from it's website(ftp://ftp.swpc.noaa.gov/pub/lists/iono_month/), and got the final ionospheric parameters. All these products had the quality identify, we chose the good quality data. For the ionosonder products from the China Meridian Project stations, we had all the raw data, so we scaled ionograms manually.

7. State the maximum tangent point – ionosonde separation distance used in NmF2 validations.

Response:

The criteria of the matching pairs of GNOS and ionosoder data have given in section 2 of the old manuscript:"The criteria used for matching GNOS and ionosonde data were a time interval within $\pm 1$ hour, and geographic latitude and longitude within $\pm 2°$ ". So, we used the geographic coordinates, the furthest distance between the maximum tangent point and ionosonde is around 200 km.

8. Since occultation hmf2 is being used for analysis in this study, it should be validated as well. It shouldn't be too difficult to compare occultation and ionosonde hmf2, similar to the Nmf2 validation.

Response:

Correct. In fact, we have validated the occultation hmF2 and shown well with about 5km bias and 25km STD between two measurements. However, considering the length of our manuscript and the main purpose of our paper aiming at analyzing the variation of the daily and zonal average of the NmF2 during the geomagnetic storm, we would like to omit the corresponding figures for validating GNOS hmF2.

9. The Discussion in Section 3.2 seems to imply that the variations in NmF2 are a geographical effect (e.g. Line 28: "large increases in the South Atlantic region"), however these are observations over a 10 hour period, and thus temporal effects would be large, particularly during a geomagnetic storm. I'm not convinced that a few occultation events over a 10 hour period are sufficient to characterize the ionospheric behavior in a particular geographical region. From Figures 7a-b, the most I would conclude is that equatorial/low latitude NmF2 increases during daytime, and decreases at night. I have the same concerns for the geographical trends are also discussed on Page 7, Lines 17-23.

Response:

Sorry, here it seems easy to lead to some misunderstanding. We noted the LT distribution of FY3C orbits flying over magnetic inclination -80o $\sim$ 80 o in day and night time is mainly around 10:00 LT and 22:00 LT as shown in figure 4 and 5. So the GNOS observations during pre-storm and storm over South Atlantic region have the similar LT in 2-3 consecutive orbits, but their events are really lack with about ten times. In other words, the GNOS observations over the same region have the similar LT distributions, therefore over the same place their differences obtained by GNOS observations between pre-storm and storm could mainly be attributed to the storm effects, although the GNOS observations are only a few times. As for the variations of NmF2 at low latitudes, in previous studies Astefyeva et al. [2015] have pointed out the most dramatic positive ionospheric storm occurred at low latitude Eastern Pacific and American

regions in the morning and post-sunset sectors. But seen from Figures 7a-b, we are difficult to evaluate the ionospheric storm effect due to the sparse GNOS observations over many regions in low latitude. As for the Eastern Pacific region between the longitudes from -180 o to -120 o over low latitude in the morning as shown in Figure 7a, there seems to be more daytime GNOS observations on 14 March 2015 during prestorm and on 18 March 2015 during recovery phase of the storm. By comparing their intensities of NmF2, we found the increase of NmF2 over this specific region, which, as one positive ionospheric storm effect, has been reported by Astefyeva et al. [2015]. Similarly, by comparing the nighttime GNOS observations over Eastern Asian in low latitude on 14 and 18 March 2015 as shown in Figure 7b, we reconfirmed the negative ionospheric storm effect which was also displayed in previous studies [e.g., Astefyeva et al., 2015; Nava et al., 2016]. About this aspect, we have added some discussion. Please see them in Section 4, Para 2-3, pp7-8.

10. Figure 8 averages NmF2/hmF2 from mid-latitude, trough, and auroral regions. Since the ionospheric structure can vary significantly over these regions, please comment on the potential effects of this averaging, and whether the trends shown in Figure 8 would change if only mid-latitudes or auroral regions were considered.

Response:

Right. The regions of GNOS observations in geomagnetic inclination 40 -80 degree would mainly locate at the mid-latitudes, and some parts would locate at main ionospheric trough, even at auroral regions, because the March 17 storm can vary significantly the ionospheric structure. As we know the trough mainly occurs in nighttime and locates just at the outside of the auroral oval; its minimum is usually situated in 60-65° magnetic latitudes in quiet time [e.g., Karpachev et al, 2016]. Along with the trough moving into the lower magnetic latitudes during the storm, the numbers of GNOS observations over the trough would increase but the corresponding NmF2 values would decrease. Thus, the zonal averaging value in NmF2 would result in decrease. However, the reverse effect on the zonal averaging value in NmF2 would be occurrence
when the auroral regions with significantly enhanced ionization in the nighttime expand largely on the equatorward during the storm [e.g., Watson et al., 2011]. Hence, these two effects on the NmF2 may counteract each other. Nevertherless, it is difficult for us to identify how many GNOS observations lie in the trough or auroral regions during the storm and then check out the exact effects on the zonal averaging in NmF2 over trough or auroral regions. Roughly, the trends shown in Figure 8 would change more or less due to the effects of these regions, but not large since most of the GNOS observations in Fig.8 located at mid-latitudes. Please see them in Section 4. Para. 3, line 15-29, pp8.

11. On a related note, please comment on the occurrence magnetospheric substorm activity, which can result in significantly enhanced ionization in the nighttime auroral region. I would strongly suggest analyzing mid-latitude and auroral regions separately, instead of a broad region covering 40 to 80 degrees inclination.

Response:

It's a good suggestion for analyzing mid-latitude and auroral regions separately. We would have studied the mid-latitude and auroral regions, individually. Unfortunately, the numbers of iononsphere occultation events were few, in Fig.8, the lowest number of the GPS occultation event was just 12 per night. In order to remain the significance of statistic, we had to analyze them in the bigger region. In the next, we plan to analyze the characteristics of this storm in smaller regions using both FY3C and COSMIC data.

12. The standard deviation for each averaged HmF2 and NmF2 value should be shown in Figures 8 and 12, perhaps as error bars.

Response:

Good. They were added in both Fig.8 and 12 in new manuscript.

13. P6 L8: NmF2 is maximum on March 16 according to Figure 8, as opposed to March 17 as stated in the text.

Response:

Here nighttime hmF2 is maximum on March 17 (green line), and nighttime NmF2 is maximum on March 16 (blue line) in Figure 8. See it in Section 3.3, line 16, pp6.

14. P6 L32: Is there an explanation for the few stations that observed a sustained daytime NmF2 enhancement on March 18?

Response:

The exact reasons for the daytime NmF2 enhancement on March 18 are not clear. The major factors are including the neutral composition changes, theremospheric winds, disturbance dynamo electric field, or some combination of those [e.g., Crowley et al, 2006; Balan, 2011; Richmond and Lu, 2000]. The further work combined with more data analysis and modeling calculations will be needed to study the mechanism of the observed results. The corresponding discussions are given in Section 4, Para. 4, lines 37 -6, pp8-9.

15. Integrating NmF2 over all local times in the top panel of Figure 12 seems meaningless.

Response:

OK. We have removed the top panel of figure 12 and re-plot the figure 12 with error bars.

16. P7 L5-6: GNOS observations at 40-80 magnetic inclination extend into the auroral region, well poleward of the northernmost ionosonde station (Moscow) in Figure 9, and thus the averaged ionosonde NmF2 wouldn't include significant auroral region effects. This may help explain discrepancies in Figure 12. For completeness, consider including ionosonde measurements from stations north of Moscow, of which there are several.

Response:

Yes. The observed regions are larger about 8 degree inclination by GNOS instrument than by ionosondes. If the more ionosonde data from stations north of Moscow are available, it will be better for us to make the comparison between two different data. Unfortunately, we have achieved none. Currently, we have recalculated the averaging NmF2 observed by GNOS within 40-72° inclination in the Supplement Figure (S. Fig.) 1. Here, the chosen 72° inclination aims at including all 17 ionosonde stations. Comparing the extents of GNOS NmF2 close to ionosonde observations between S. Fig. 1 and Figure 12, we can see that the approaching is better in S. Fig. 1 than in Figure 12, especially in the daytime. In the S. Fig. 1, the discrepancies between GNOS and ionosonde also exist. There should be other factors resulting in the obvious discrepancies. The major factor is due to the GNOS observations outspreading over the world, whereas the ionosondes locate at the several longitudes on the continents.

Supplement Figure 1: Comparison of averaging NmF2 values from the 17 ionosonde stations and GNOS in the 40-72° inclination in the NH.

17. P7 L6-7: "..indicating significant differences still exist between the two measurement techniques." Please specify the differences this statement is referring to.

Response:

Good. These discrepancies may be due to the significant differences of the spatial observations between the two measurement techniques, i.e., GNOS can observe globally, whereas ionosondes mainly locate at several longitudes on the continents. We have made a revision in the manuscript accordingly. Please see it in Section 3.4, Para. 2, lines 14-16.

18. P7 L13: Spherical homogeneity of ionospheric density is also a very large assumption at high latitudes (trough, auroral, polar cap regions).

Response:

Yes. We have stated it in the section 4 , Page 7, line 23.

19. P8 L7: Instead of "GNOS data", specify that GNOS NmF2 values are reliable, since this was the only parameter validated in the manuscript.

Response:

Ok, it a scrupulous expression, we accept it and corrected in new manuscript, Page 9, line 19.

Technical Corrections:

1. P1 L16: I suggest "space detection" be replaced with "space-based remote sensing" or something along those lines.

Response:

Ok, we replaced it in Page 1, line 16 in new manuscript.

2. P1 L26-28: Unless I'm misinterpreting the intended meaning, this sentence should read something like: "In the zone of 40-80° magnetic inclination, average NmF2 observed by GNOS and 17 ground-based ionosondes showed the same basic trends during the geomagnetic storm.

Response:

Yes, we rewritten this sentence, and make it more accessible, see in Page 1, line 27-29 in new manuscript.

3. P2 L4: "The GNSS radio occultation technique uses occultation receivers..."

Response:

OK. We revised it, see in Page 2 , line 4.

4. Figure 1 axis labels are difficult to read.

Response:

Sorry. We have revised it and replaced a new figure 1 in new manuscript.

We appreciate for your warm work earnestly, and hope that the response will meet with approval.

The references and the new update version paper are enclosed in supplement.

Thank you very much.

Yours sincerely,

Bai Weihua, Wang Guojun and other co-authors

Please also note the supplement to this comment:
http://www.atmos-meas-tech-discuss.net/amt-2016-291/amt-2016-291-AC2-supplement.zip
* * *
[Figure]

[Figure]

**Fig. 1.** Comparison of averaging NmF2 values from the 17 ionosonde stations and GNOS in the 40-72° inclination in the NH.